# Perch Positioning Affects both Laying Hen Locomotion and Forces Experienced at the Keel

**DOI:** 10.3390/ani10071223

**Published:** 2020-07-18

**Authors:** Christina Rufener, Ana K. Rentsch, Ariane Stratmann, Michael J. Toscano

**Affiliations:** 1Center for Animal Welfare, Department of Animal Science, University of California, Davis, CA 95616, USA; 2Center for Proper Housing: Poultry and Rabbits, Animal Welfare Division, University of Bern, 3052 Zollikofen, Switzerland; arentsch@uoguelph.ca (A.K.R.); ariane.stratmann@vetsuisse.unibe.ch (A.S.); michael.toscano@vetsuisse.unibe.ch (M.J.T.); 3Department of Animal Biosciences, University of Guelph, Guelph, ON N1G 2W1, Canada

**Keywords:** perch, position, angle, distance, acceleration, peak force, locomotion

## Abstract

**Simple Summary:**

Keel bone fractures in laying hens can occur due to falls and collisions within the housing system, although other factors such as genetics and nutrition contribute to the high fracture prevalence found in commercial laying hens. In addition, routine behaviors such as dustbathing or locomotion might contribute to the problem due to accumulated forces at the keel. To understand how locomotion affects the risk to sustain a fracture, we trained 20 brown and 20 white laying hens to jump from a platform to a perch installed at different angles, distances, and directions. We found that longer distances and steeper angles—especially during downward transitions—resulted in higher force at the keel and were more difficult for the hens to navigate. Our results show that perch position has an impact on the forces which a keel bone needs to absorb during controlled movements. In addition, perch position affects the hens’ ability to move safely from perch to perch, i.e., without falls and collisions. Optimizing perch position could help to create a safer environment for laying hens and might reduce keel bone fractures.

**Abstract:**

The aim of this study was to assess the effect of perch positioning on laying hens’ locomotion and the resulting energy experienced at the keel. Twenty Nick Chick and 20 Brown Nick hens were trained to transition from a platform to a perch in several configurations. Three variables of perch positioning were tested in a 2 × 2 × 2 factorial design: direction (upward vs. downward), angle (flat vs. steep), and distance (50 cm vs. 100 cm). All hens were tested for five jumps of each treatment combination at 27–28 weeks of age. As predicted, we found steep angles and long distances to result in higher peak forces and impulse during take-off, flight, and landing; longer latency to jump; a higher likelihood to perform balancing movements; and a longer latency to peck at the provided food reward. The effect of perch positioning on locomotion and force at the keel during downwards jumps and flight was more pronounced in Brown Nick hens than in Nick Chick hens. Although we cannot state how the observed forces at the keel relate to the risk for keel bone fractures, our results indicated that optimizing perch positioning can reduce accumulated forced at the keel and consequent risk for fracture due to unsuccessful transitions.

## 1. Introduction

Alternative housing systems for laying hens such as aviaries are becoming more common as consumer demand shifts towards eggs from non-cage systems [1]. Although aviaries provide many benefits to hens, e.g., the ability to perform species-specific behaviors [2,3], it is speculated that the height and difficulty maneuvering within the system increase the risk for falls and injuries [4,5]. Falls can result in collisions with other hens or pen furnishings which are assumed to be one of the reasons for the high prevalence of keel bone fractures in laying hens (reviewed in [6]). The risk for falls and collisions likely relates to positioning of furnishings that determine how birds move between them. Characteristics of movements, including direction [7,8], angle [8,9,10], and distance [11,12], affect whether hens can move successfully between perches.

Although falls and collisions are suspected to result in fractures [13], a single histopathological characterization of keel bone fractures indicated that high energy collisions are unlikely to be the underlying cause for the majority of fractures in laying hens [14]. An alternative pathogenesis to trauma might be accumulated forces during routine behaviors such as roosting or dust bathing (reviewed in [6]). Similarly, stress fractures in humans occur when bone is exposed to repeated stress (fatigue fractures [15]) or due to stress applied to a bone with deficient elastic resistance (insufficiency fractures [16]). Thus, fractures may conceivably result from both high as well as low energy forces applied to the keel necessitating a need for a comprehensive understanding of how collisions as well as controlled movements relate to forces experienced at the keel. 

Although there is evidence that increasing perch height is associated with increasing forces acting on the hen itself [17], it is unknown how these forces are experienced at the keel during innocuous routine activities. The aim of the current study was to assess the effect of perch positioning on bird locomotion and the resulting energy experienced by the keel. Multiple perch positions similar to those used in commercial aviary systems were tested for their effect on hen locomotion-related behaviors and physical properties of movement. Peak force and impulse at the keel during take-off, flight, and landing were measured by an accelerometer placed in a fabric vest with the sensor directly on the keel bone, 3 cm above the caudal tip. We hypothesized that angle, distance, and direction of movement would affect forces experienced at the keel and navigation-related behaviors (e.g., latency to jump and balancing movements). 

## 2. Animals, Material and Methods

### 2.1. Ethical Approval

Ethical approval to conduct the study was obtained from the Veterinary Office of the Canton of Bern in Switzerland (approval number BE22/17). The experiment complied with Swiss regulations regarding the treatment of experimental animals.

### 2.2. Animals and Housing

Eighty Nick Chick and 80 Brown Nick (white and brown feather colored, respectively) day-old chicks provided by a commercial hatchery were reared in one pen of a barn alongside 440 other chicks (300 Nick Chick, 300 Brown Nick in total). The rearing pen contained a commercial multi-tier aviary system (NATURA Aufzucht 187, Inauen AG, Appenzell, Switzerland) with two grid and two perch tiers, a floor covered with wood shavings, and an enclosed wintergarden with perches that the pullets had daily access to from six weeks of age. The rearing system provided access to two tiers with round metal perches coated with a 2 mm layer of plastic (outer diameter: 3.6 cm) installed at 43 cm above the first grid tier and 50 cm above the second grid tier. The total height of the aviary was 195 cm. Birds moved between grid tiers using platforms that were installed at both aviary sides. In the wintergarden, wooden, rectangular A-frame perches at 54 and 110 cm height with an angle of 58° between them were available. Hens were given ad libitum access to a standard pullet diet. Stocking density during rearing was based on the Swiss Animal Welfare Ordinance with 16.4 birds per m^2^ grid surface. 

At 16 weeks of age, hens were transferred to an experimental barn and randomly assigned to eight identical pens (n = 20 birds per pen; four pens with Nick Chick and four pens with Brown Nick hens). Each hen was identified by a unique number on a plastic, flexible leg band (Roxan ID, Selkirk, UK) on the left leg and a specific combination of colored leg rings referring to the pen-specific identification number on the right leg. 

The eight pens were set up side by side in the middle of an experimental barn. The distance between the building wall and the short side of the pens was approximately 2 m. Each pen included two compartments: a home pen and a test pen (Figure 1). The home pen (4 m × 2 m) contained a 7.4 m^2^ littered area, a round feeder (diameter: 50 cm), five nipple drinkers, and a nest box (area: 0.6 m^2^). Two round metal perches (1.5 m, 3.2 cm diameter) and an elevated metal grid platform (width × length: 30 cm × 1.5 m) were installed at a height of 1.3 m. Perches were arranged on either side and parallel to the platform separated by a distance of 30 cm. Perches and platform were placed at the same height in order to prevent bias developing in the new environment in terms of jumping abilities, i.e., hens being familiar with a certain angle or distance. Hens were discouraged from jumping to these elements directly from the ground by the combination of structure height (1.3 m) and pen width (2 m) resulting in a steep angle and long diagonal distance between floor and elevated structures. Instead, a ramp was installed which transitioned to the platform from the floor (length: 1.78 m; angle from the floor: 47°) to make the platform and perches accessible. Besides preventing the hens from developing biases regarding specific angles or distances, the arrangement of platform and perches ensured that hens were familiar with all surfaces used in the test procedure. As the elevated elements were accessible via the ramp, hens walked upwards on the ramp until they reached the platform with perches positioned parallel. Most hens also used the ramp to descend from the perches, though some individuals jumped from the perches to the litter directly (personal observations). The platform/perch arrangement simulated the conditions that were tested in the experiment, where hens were required to jump from a platform and land on a perch.

The test pen (area: 2 m × 2 m) contained two vertical poles which held the landing perch during testing. In between training sessions, all testing equipment was removed from the test pen which was then made freely accessible to all hens. The test pen also contained litter to the same depth as the home pen. The grid of the wall and the door separating test pens and home pens was covered with plastic sheets to inhibit visual contact between pens during training and testing.

Light was provided by LED lamps from 04:00 h with a 10 min dawn phase to 19:00 h with a 25 min dusk phase. Windows were incorporated in the walls of the building, i.e., approximately 2 m away from the short sides of the pens, and availability of daylight was controlled with curtains which were raised between 05:00 and 18:00 h. Hens were provided feed ad libitum with a standard layer diet (FORS 2051, Burgdorf, Switzerland). Animal care staff always entered the pens through the door of the home pen, whereas the experimenters entered the pens through the door of the test pen.

### 2.3. Rewarding Apparatus 

Hens were trained using a rewarding apparatus that was constructed from a commercial feed trough and covered with a metal plate. Using a switch connected to the device by a 2 m cable, a flap over the metal plate covered the food reward (wheat grains) but could be opened and closed. The rewarding apparatus was intended to standardize the rewarding process and prevent hens from being distracted by the experimenter. The apparatus used signals (LED lighting and sounds) to communicate the trial’s initiation and availability of food. For habituation to the rewarding apparatus, a dummy rewarding apparatus was placed in all eight test pens from the first day of population on (16 weeks of age). The dummy rewarding apparatus was the same size and material as the rewarding apparatus, but the flap covering the reward was constantly open.

### 2.4. Habituation to the Test Pen and Selection of Focal Hens

After a week of habituation to the new environment with both test pen and home pen being freely accessible, hens were gradually introduced to isolation in the test pen over a two-week period. Initially, hens were isolated in groups of two (one week of daily, 10 min sessions) and then individually (one week of daily, 5 min sessions). Observations of the hens’ behavior (e.g., escape attempts or stress calls) were conducted and used to exclude hens which did not habituate to isolation in the test pens. Hens that were excluded from the study were not assessed further. Within the same initial three-week period (one week of habituation and two weeks of isolation), hens were also introduced to the dummy rewarding apparatus containing freely available wheat grains. At 19 weeks of age, the dummy was gradually replaced with the actual test apparatus. Depending on the hens’ individual interaction with the test apparatus, 40 Brown Nick and 40 Nick Chick hens were selected as focal hens. A full description of the habituation protocol and hen selection criteria is available in Appendix B.

### 2.5. Training

From 20 weeks of age, selected focal hens (40 Brown Nick, 40 Nick Chick hens) were introduced to the testing apparatus in the test pen (Figure 2). Vertical poles in the test pen held the landing perch and rewarding apparatus. For take-off, a platform identical to the platform in the home pen was used. The landing perch and the rewarding apparatus could be adjusted vertically, whereas the take-off platform could be moved horizontally only. In order to create all test conditions, two take-off platforms of different heights (60 cm above ground for upward jumps and 135 cm above ground for downward jumps) and variable heights of the landing perch were combined. 

Hens were trained individually on five days per week between 8:00 and 12:30 h. The order of pens was stratified to expose hens to training at different times of day. No feed restriction prior to training was applied. Before training started the doors between the test pen and home pen were closed in all pens, thus a hen being trained in the test pen did not have visual contact to hens in the home or adjacent test pens. Within pens, focal hens were picked for training randomly.

Training consisted of stages with each lasting one week in duration. After each week, focal hens not fulfilling inclusion criteria were excluded from further training. Whereas the first two weeks of training served to habituate the hens to the testing apparatus in the test pen, the next four weeks aimed to prepare the hens for the test conditions applied during the testing phase. During the last stage of training, remaining hens were habituated to a vest of flexible fabric (previously used in [18,19]) that allowed full freedom of movement and contained the accelerometer. Vests were custom made with two different sizes and colors (white and brown, respectively) for Nick Chick and Brown Nick hens. The vests allowed the accelerometer to be placed directly at the keel during testing as well as a number on the back for identification. A full training protocol is provided in Appendix C.

### 2.6. Experimental Design and Test Conditions

Twenty Nick Chick and 20 Brown Nick hens (4–7 hens/pen) completed the training protocol successfully and were included in the study. Three factors of perch positioning were tested in a 2 × 2 × 2 factorial design (i.e., 8 treatment combinations or conditions; Figure 3): direction (upward vs. downward), angle (flat: 30° for upward, 15° for downward jumps vs. steep: 60° for upward, 30° for downward jumps), and diagonal distance (50 cm vs. 100 cm). All hens were tested for all conditions. Angles and distances were selected based on existing literature [7,9,11] to include a range of difficulty but unlikely to result in collisions. 

Due to the time required for the installation of the testing equipment, half of the pens (10 Nick Chick and 10 Brown Nick hens) were tested per day. Testing took place on five days per week from 8:00 – 12:00 h over a period of 17 days. Pens and hens within pens were tested in a predefined order, whereas the order of the conditions was counterbalanced between hybrids, individuals, and time of day. As 40 hens were tested for eight conditions each, 320 test sessions were conducted on 16 days (27–30 weeks of age). Per test session, hens were required to jump 3–5 times depending on the motivation to jump, resulting in approximately 1600 jumps for the experiment.

### 2.7. Data Collection

The doors between the test and home pens were closed in all pens during testing. Within the test pen, a video camera was connected to a laptop with a screen displaying a time stamp (described below). Vertical height of the landing perch and rewarding apparatus and horizontal distance of the take-off platform was adjusted for the first hen/condition. The hen was caught and checked for the absence of injuries, wounds, and bumble foot, and palpated to detect the presence or absence of keel bone fractures [20], but no such conditions were found during the testing phase. The accelerometer was placed in the hen’s vest, the hen placed on the landing perch, the video recording started, and the flap of the rewarding apparatus opened. After five seconds of feeding, the flap of the rewarding apparatus was closed, and the hen returned to the take-off platform with the experimenter’s hand remaining on the back of the hen until initiation of the signal. After a successful jump, the hen was allowed to feed for five seconds. The procedure was repeated five times with the first serving to acclimate the hen to the procedure. After the final transition, the accelerometer was removed and the hen returned to the home pen.

#### 2.7.1. Video Recording

Test sessions were recorded with 240 frames per second (fps) using a high-frequency camera (GoPro HERO5 black, GoPro, Inc., San Mateo CA, USA). The camera was installed at the metal grid wall dividing the test pens. As the camera provided an internal time stamp only which could not be synchronized with the other testing equipment, a screen (EIZO FORIS FG2421; EIZO, Hakusan, Ishikawa, Japan) showing the computer time with a frequency of 120 fps was installed in the test pen during testing. 

Using video software enabling movement between single frames (SMPlayer), the duration of the variables described in Table 1 were assessed with a resolution of milliseconds (0.000 s) using an adjustment described in Appendix C. For each transition, multiple variables were calculated including latency to jump (time from Signal to Take-off start; Table 1), duration of take-off (time from Take-off start to Take-off end), duration of flight (time from Take-off end to Landing start), duration of landing (time from Landing start to Landing end), and the latency to peck (time from Landing end to First peck). The occurrence of balancing movements after landing (i.e., first contact with the landing perch) was also recorded. Balancing movements (recorded separately but summarized as a binary variable) were defined as wing flaps using both wings or asymmetrically with one wing, sideway steps, or rotation of the body (back, forth, or sideways). Safe landings were defined as landings not resulting in falls in collisions [13], and could include landings followed by balancing movements or asymmetric landings. Unsuccessful landings, i.e., landings resulting in falls or collisions, were recorded. Very few landings during the testing safe were classified as unsuccessful and thus, these data were not presented here.

#### 2.7.2. Acceleration Sensors and Calculations

The acceleration during jumps was measured using a triaxial accelerometer (MSR165 data logger, custom-made to fit in case MSR B16; size: 27 × 16 × 53 mm, weight: 27 g; MSR Electronics GmbH, Seuzach, Switzerland) with an external acceleration sensor (cable length: 20 cm). The accelerometers were synchronized with video recordings and fit in a pocket at the back of a vest worn by the hens. The external sensor was placed directly on the keel (approximately 3 cm above the caudal tip) via the cable that was laid underneath the fabric of the vest from the back along the crop and breast. Acceleration was measured for x-, y-, and z-axis with a frequency of 800 Hz (800 measurements / sec) and a maximum sensitivity of ±200 g.

Acceleration data were transferred to the computer via dedicated software (MSR version 6.01.00) and saved as CSV-files. From the axis-specific acceleration, the combined magnitude of acceleration, i.e., the total acceleration acting on the hen was calculated:(1)$$acceleration\;\left[ms^{-2}\right]=\sqrt{a_{x}^{2}+a_{y}^{2}+a_{z}^{2}},$$
with $a_{x}$, $a_{y}$, and $a_{z}$ describing acceleration in the *x*, *y*, and *z* direction, respectively. Force was calculated using Newton’s second law: (2)F=ma,
where m was the body mass [kg] of the hen. Body mass was recorded with an electronic scale before and after the 16 days of testing and the average used for calculation. Peak forces [N] during take-off, flight and landing within the time frames extracted from video recordings (Table 1) were identified using R, version 3.4.2 [21].

In order to describe the accumulated forces to which a keel was exposed, the impulse for take-off, flight and landing was calculated by multiplying the average force of each phase with the respective duration: (3)$$impulse\left[Ns\right]=\bar{F}\Delta t.$$

### 2.8. Statistical Analysis

Statistical analysis was conducted in R, version 3.4.2 [21], using linear mixed-effect models (LMER) and generalized linear mixed-effect models (GLMER) with package ‘lme4’ [22]. Model assumptions regarding normality of errors and homoscedasticity were checked by graphical analysis of residuals. For GLMER models, residuals were simulated using package ‘DHARMa’ [23]. Data were log or inverse (1/*x*, inv) transformed if necessary. The final models were obtained by a stepwise-backwards reduction of the full model. Parametric bootstrap tests with package ‘pbkrtest’ [24] were used for model comparison with a *p*-value of >0.05 as a criterion of exclusion. The package ‘effects’ [25] was used to calculate and display model estimates.

Statistical analysis was conducted in R, version 3.4.2 [21], using linear mixed-effect models (LMER) and generalized linear mixed-effect models (GLMER) with package ‘lme4’ [22]. Model assumptions regarding normality of errors and homoscedasticity were checked by graphical analysis of residuals. For GLMER models, residuals were simulated using package ‘DHARMa’ [23]. Data were log or inverse (1/*x*, inv) transformed if necessary. The final models were obtained by a stepwise-backwards reduction of the full model. Parametric bootstrap tests with package ‘pbkrtest’ [24] were used for model comparison with a *p*-value of >0.05 as a criterion of exclusion. The package ‘effects’ [25] was used to calculate and display model estimates.

Data were analyzed separately for upwards and downwards jumps as conditions were not comparable directly due to different angles depending on jump direction (flat: 30° up vs. 15° down, steep: 60° up vs. 30° down). Response variables for LMER (and the applied transformation) were peak force during take-off (inv), flight (inv), and landing (inv); impulse during take-off (up: untransformed, down: log) and landing (log); latency to jump (log); and latency to peck (log). Duration of flight was highly correlated with distance and thus, impulse was only tested for take-off and landing. The likelihood for balancing movements (binary variable) was evaluated using GLMER. Fixed effects included in all models were distance (50 cm, 100 cm), angle (flat, steep), hybrid (Nick Chick, Brown Nick), and all interactions. Jump number nested in condition (factor with 4 levels: 50 cm flat, 50 cm steep, 100 cm flat, 100 cm steep) nested in hen nested in pen was used as a random effect to account for individual and pen differences and to prevent pseudo-replication. Including jump number in the random effect considered learning effects or adaptation to the specific condition within one test session. Calendar date was included as a crossed random effect to account for management and environmental effects, e.g., temperature.

## 3. Results

An overview of the results is presented in Table 2. The full data set is available in the Appendix A.

### 3.1. Peak Force at the Keel

#### 3.1.1. Upward Transitions

Peak force during take-off for upward transitions of 100 cm distance was greater than during take-off for upward transitions of 50 cm distance (estimated means (estimated 95 % confidence interval): 50 cm = 3.17 (2.84, 3.59) N, 100 cm = 3.61 (3.19, 4.17) N; *p* = 0.027).

Peak force during flight was linked to an interaction of distance * angle (*p* = 0.037). Peak force during flights of 50 cm distance was greater if the angle was flat compared to steep (flat = 3.21 (2.86, 3.66) N, steep = 2.72 (2.46, 3.04) N), whereas during flights of 100 cm distance, there was no difference in peak force between flat and steep angles (flat = 3.42 (3.03, 3.94) N, steep = 3.51 (3.08, 4.06) N). 

Peak force during landing was greater after transitions upwards of 100 cm than 50 cm upward transitions (50 cm = 2.93 (2.64, 3.29) N, 100 cm = 3.22 (2.87, 3.66) N; *p* = 0.019).

#### 3.1.2. Downward Transition

Peak force during take-off for downward transitions was greater in Brown Nick hens than Nick Chick hens (Nick Chick = 3.39 (3.02, 3.87) N, Brown Nick = 4.04 (3.52, 4.76); *p* = 0.032).

The same pattern emerged during downward flight with Brown Nick hens experiencing greater peak force than Nick Chick hens (Nick Chick = 3.22 (2.88, 3.66) N, Brown Nick = 3.95 (3.44, 4.65) N; *p* = 0.008). In addition, peak force during downward flight was associated with a distance * angle interaction (*p* = 0.038). Although downward flight of 50 cm distance resulted in peak forces similar for flat and steep angles (flat = 3.32 (2.91, 3.88) N, steep = 3.19 (2.81, 3.69) N), downward flight of 100 cm was associated with a greater peak force if the angle was steep in comparison to a flat angle (flat = 3.45 (2.99, 4.06), steep = 4.31 (3.64, 5.30) N).

Peak force during landing was greater after downward transitions of 100 cm than 50 cm (50 cm = 2.92 (2.66, 3.24) N, 100 cm = 3.22 (2.90, 3.61) N; *p* = 0.019). Further, peak force during downward landing was associated with a hybrid * angle interaction of (*p* = 0.026). Peak force in landing Nick Chick hens was independent of angle (flat = 2.84 (2.54, 3.24) N, steep = 2.89 (2.57, 3.31) N) whereas in Brown Nick hens peak force during landing was greater after steep transitions compared to flat (flat = 3.23 (2.85. 3.75) N, steep = 3.31 (2.91, 3.86) N).

### 3.2. Impulse 

#### 3.2.1. Upward Transition

Impulse (accumulated forces at the keel) during upward take-off was associated with a distance * angle interaction (*p* = 0.036). During take-off for transitions of 100 cm, impulse was slightly lower in flat than steep angles (flat = 0.29 (0.25, 0.33) Ns, steep = 0.27 (0.22, 0.31) Ns). During take-off for transitions of 50 cm, flat angles were linked to a greater impulse than steep (flat = 0.30 (0.26, 0.34) Ns, steep = 0.27 (0.22, 0.31) Ns). 

Impulse during landing was linked with an interaction of distance * angle (*p* = 0.001). Landing after 50 cm distance transitions resulted in a similar impulse for flat and steep angles (flat = 1.41 (1.23, 1.62) Ns, steep = 1.42 (1.23, 1.64) Ns). Impulse during landing after 100 cm distance transitions was higher than during landing after 50 cm transitions and increased in steep angles compared to flat angles (flat = 1.60 (1.339, 1.85) Ns, steep = 2.32 (2.01, 2.68) Ns). 

#### 3.2.2. Downward Transition

Impulse during downward take-off was higher in Brown Nick hens (0.53 (0.46, 0.61) Ns) than in Nick Chick hens (0.39 (0.34, 0.45) Ns; *p* = 0.018). We further found a distance * angle interaction relating to impulse during downward take-off (*p* = 0.004). Impulse during take-off for transitions of 50 cm was similar for flat and steep angles (flat = 0.45 (0.40, 0.52) Ns, steep = 0.47 (0.41, 0.54) Ns), whereas transitions of 100 cm and steep angles resulted in higher impulse than transitions of 100 cm and flat angles (flat = 0.36 (0.32, 0.42) Ns, steep = 0.51 (0.45, 0.59) Ns).

During landing after downward transitions, impulse was higher after transitions of 100 cm compared to transitions of 50 cm (50 cm = 1.59 (1.37, 1.86) Ns, 100 cm = 2.04 (1.75, 2.38) Ns; *p* = 0.001, Figure 4).

### 3.3. Latency to Transition

#### 3.3.1. Upward Transition

Latency to transition between the platform and perch was greater in Brown Nick hens than Nick Chick hens (Nick Chick = 1.38 (1.07, 1.80) sec, Brown Nick = 2.78 (2.16, 3.59) sec; *p* = 0.016). Latency to transition was also associated with a distance * angle interaction (*p* = 0.048, Figure 5). Latency to transition upward was similar for flat and steep angles of 50 cm (flat = 1.44 (1.16, 1.78) sec, steep = 1.38 (1.11, 1.71) sec). Latency to transition 100 cm was greater than a distance of 50 cm, whereas steep angles for 100 cm upward transitions were associated with increased latency compared to flat angles (flat = 2.61 (2.10, 3.24) sec, steep = 3.32 (2.66, 4.15) sec).

#### 3.3.2. Downward Transition

Latency to transition downward was higher for 100 cm transitions (3.89 (3.11, 4.87) sec) than 50 cm (2.31 (1.85, 2.89) sec; *p* = 0.001). Latency to transition downward was further linked to angle (*p* = 0.001), with a greater latency to transition with steep (3.41 (2.72, 4.26) sec) than flat angles (2.58 (2.06, 3.23) sec).

### 3.4. Balancing Movements at Landing

#### 3.4.1. Upward Transition

The likelihood to perform balancing movements was associated with a distance * angle interaction (*p* = 0.0001; Figure 6). Hens were similarly likely to perform balancing movements after 50 cm distance transitions of flat and steep angles (flat = 23.2 (16.0, 32.5) %, steep = 17.0 (11.0, 25.2) %) as well as 100 cm distance transitions of flat angles (21.5 (14.4, 30.9) %), whereas the likelihood to perform balancing movements was greater after transitions characterized by 100 cm and steep angles (57.9 (45.9, 68.9) %).

#### 3.4.2. Downward Transition

Hens were more likely to perform balancing movements after downward transitions of 100 cm than after 50 cm (50 cm = 35.6 (30.2, 41.3) %, 100 cm = 53.9 (47.8, 60.0) %; *p* = 0.0001).

### 3.5. Latency to First Peck

#### 3.5.1. Upward Transition

After upward transitions, the latency to peck was linked to a distance * angle interaction (*p* = 0.001; Figure 7) with similar values after 50 cm flat and steep angle transitions (flat = 0.97 (0.89, 1.07) sec, steep = 1.01 (0.92, 1.11) sec) and 100 cm flat transitions (1.05 (0.95, 1.15) sec). In contrast, the latency to peck was greater after 100 cm steep transitions (1.31 (1.19, 1.45) sec).

#### 3.5.2. Downward Transition

Latency to peck after downward landing was greater in Nick Chick than Brown Nick hens (Nick Chick = 1.11 (1.02, 1.20) sec, Brown Nick = 0.94 (0.86, 1.02) sec; *p* = 0.034). We found an interaction of distance * angle relating to the latency to peck after downward transitions (*p* = 0.006). Latency to peck was similar after downward, 50 cm transitions for flat, and steep angles (flat = 0.95 (0.88, 1.03) sec, steep = 0.92 (0.85, 0.99) sec). Latency to peck was greater after downward transitions of 100 cm, especially if the angle was steep (flat = 1.06 (0.98, 1.14) sec, steep = 1.22 (1.12, 1.32) sec).

## 4. Discussion

The aim of this study was to assess the effect of perch positioning on laying hen locomotion and the resulting energy experienced at the keel. As predicted, we found that longer distances and steeper angles—especially during downward transitions—resulted in an increased peak force and impulse at the keel and were more difficult for the hens to navigate based on longer latency to transition, higher likelihood for balancing at landing, and a higher latency to peck at the reward after landing. Our results indicate that distance and angle between take-off and landing structure could contribute to the high keel bone fracture prevalence observed in complex aviary systems. Accumulated forces during routine behaviors and the risk for high-impact collisions due to difficulties in transitions between perches might be the two main mechanisms associated with perch positioning and its relation to keel bone fracture risk.

During observations of flight and landing, transitions that were characterized as longer and steeper in either the up or down orientation appeared to be more difficult for hens as indicated by the greater latency to transition and the behavior after contacting the perch, i.e., time to peck and balancing. Our findings generally agree with these of others [7,8,10,11]. The current study used hens that were trained and thus familiar with the testing paradigm which allows for a greater consideration of the factors that make certain transitions more difficult. Specifically, the increased latency to transition with steeper angles could suggest that hens required a greater amount of time to position themselves and cognitively process the information needed for flight including the position of the landing perch and/or thrust required for take-off. Moinard et al. [26] calculated trajectories during take-off and landing by observing the eye position of hens in experimental flight and suggested that hens were gathering visual information about the perch to determine needed take-off thrust. Latencies in downward movements in our study were approximately 0.9 s greater than upward movements, which also agrees with existing literature that the former are more demanding [7,8].

Once the hen has taken off, landing will involve a separate set of processes as the hen seeks to adjust her vertical position, speed, and extension of the feet to make contact with the landing surface [27]. The duration of balancing behaviors and latency to peck at the food reward in the current study was increased in longer and steeper transitions suggesting the greater difficulty of these movements. Moinard et al. [26] suggested the greater variation in eye position relative to the perch during landing in more challenging transitions could relate to difficulty adjusting the hen’s position during flight. Our observations once the hen has landed could be an extension of those compensating behaviors performed in the approach period as the hen continues to adjust her position on the perch. Specifically, if the head is positioned too far forward over the perch or not far enough as she makes initial contact, the hen must perform balancing movements (e.g., use her wings or shift her legs) to ensure she does not topple over. The finer elements of landing behavior may help to explain previous findings where the reduced slipperiness of different perch types was believed to improve landings [10] and require less balance movements [28].

The load experienced by the keel bone (represented by impulse and peak force during take-off, flight, and landing) was greater in magnitude during landing following longer and steeper transitions as expected. The force acting on the whole hen during landing has been shown to be considerably greater (mean peak force: 81.0–106.9 N [17]) than the peak force measured directly at the keel in the present study (2.92–4.04 N) likely because the majority of kinetic energy during a safe landing is absorbed by the feet and legs. Unsafe landings resulting in collisions will result in much greater load at the keel with accelerations of >100 G (=150 N for a hen of 1.5 kg bodyweight [18,29]), but it is not clear whether these forces are directly related to the occurrence of keel bone fractures. Future efforts will need to determine the upper threshold of fracture risk in terms of collision energy and other related factors during events resulting in trauma, e.g., collisions. One method to determine such thresholds would be an impact testing paradigm used by our group in a variety of situations [30,31,32], though conditions would need to be comparable.

Hens of the current study experienced a limited number of poor landings, and only safe transitions without falls or collisions were considered. On one hand, the relatively low energies the keel was required to absorb during controlled transitions may represent a ‘safe’ quantity of energy that the keel can experience. On the other hand, forces applied to the keel during controlled movements between perches as observed in the present study are less likely to cause trauma-related fractures, although could contribute to fatigue fractures which occur when normal bone is exposed to repeated stress [15]. Supporting this position, recent information has suggested that fractures could result from events other than trauma-related collisions (reviewed by [33]). Casey-Trott et al. [34] suggested that greenstick fractures (i.e., incomplete, bending fractures commonly found in developing bone) might result from routine behaviors, while Harlander-Matauschek et al. [6] specified the need to investigate low energy non-collision events as a cause of fracture. Evidence that many fractures lack the pathological signs of traumatic injury have been reported [14], with similar findings elsewhere [35] though the latter authors’ distinction between fractures and deviations of the keel was not clear. Taken together, these reports highlight a need to understand the nature of the forces to which the keel is exposed and how this might lead to non-traumatic fractures.

Overall, both accumulated forces as well as difficulties in transitions between perches seem to be related to angle and distance between elements of the housing system as well as the direction of movement. Our results are relevant for commercial conditions, especially aviaries, due to several reasons. First, laying hens housed in aviaries are required to use aerial locomotion in order to move vertically through their environment and to reach all resources. As a result of the spatial distribution of resources in aviary systems, hens cannot avoid applying repeated load on their keel bones when using their flight muscles [36], and thus have to expose their keel to accumulated forces and corresponding risk for stress fractures when accessing feed, water, nest boxes, perches, or litter. Whereas these forces might not be problematic in healthy hens, weakened bones due to high egg laying rates [37] or other disease conditions [33] would contribute to increased fracture susceptibility and the risk for insufficiency fractures (when stress is applied to a bone with deficient elastic resistance [16]). Second, we found that hens responded differently to perch positioning depending on strain (i.e., brown vs. white hens). Strain-specific differences in locomotion and navigation abilities within different aviary designs relate to keel fracture frequency and severity [38], therefore the design of commercially available aviary systems might need to consider the used genetic line. Third, our results were obtained from hens with unfractured keels based on palpation. Given that keel bone fractures affect hen mobility [39,40,41], the high fracture prevalence found in older birds kept in experimental and commercial systems [5] might amplify the negative effect of steep angles and long distances on transition success. Fourth, hens in the present study were carefully trained in order to transition between all perch positions successfully. Considering that angles and distances of hen transitions in commercial systems are often steeper or longer than the present study (up to 105 cm diagonal distance and 80° between tiers in Swiss aviary systems [42]), accumulated forces and the risk for trauma-related fractures is presumably higher if hens are not familiar with their environment. The transition between rearing and layer environments and the period immediately following should be considered in the development of keel bone fractures as hens are usually exposed to a higher and more complex system than they are experienced with (reviewed in [43]).

In order to reduce the load on the keel bone which could result in keel bone fracture occurrence, we recommend distances and angles for paths within a housing system not to exceed 50 cm and 30° as these perch positions have been shown to be more successfully navigated than steeper angles and longer distances in previous studies. However, distances and angles below 50 cm and 30°, respectively, did not avoid falls in all cases [8,9]. Accordingly, increased distance between rows of an aviary can be a risk factor for keel bone fractures [44]. More specific recommendations are needed for multi-level housing system such as aviaries with increased difficulty in transitioning between levels. Periods of increased local density in certain areas (i.e., upper perches immediately before and during lights out [45]) and other hens being present at the take-off or landing site are likely to decrease the chance of a successful landing [46]. In addition, dimmed light during dawn and dusk can affect transition accuracy and thus, the risk for falls [7,47] though lighting conditions only affect landing accuracy and the latency to jump when the intensity is very low (i.e., below 2 Lux [7,47]). Alternative types of inter-tier movement such as ramps have been shown to be beneficial regarding both vertical space use [48] as well as keel bone fracture development [13,49]. The specific forces hens experience using ramps and the potential benefit to promoting bone health should be evaluated.

## 5. Conclusions

Although we cannot state how the observed forces relate to the risk for keel bone fractures directly, the results of the present study indicate that perch positioning affects accumulated forces at the keel during take-off, flight and landing. Moreover, the associations between perch position and locomotion behavior (e.g., longer latency to jump, higher likelihood to perform balancing movements, and longer latency to peck after transitions between steep angles and long distances) indicate that perch positioning affects the hen’s ability to successfully navigate between perches. Our results show that optimizing perch positioning is crucial for successful transitions within the housing system and could be an approach to reduce keel bone fracture prevalence in commercial laying hens housed in aviaries.

## Figures and Tables

**Figure 1 animals-10-01223-f001:**
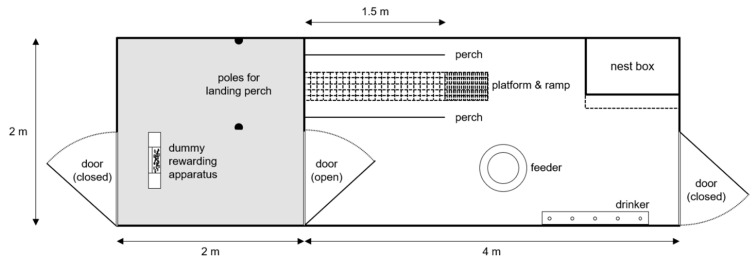
Top view of an experimental pen with test pen (left; gray background) and home pen (right) where hens had access to both pen compartments (i.e., if no training or testing took place). The home pen contained a feeder, nipple drinkers, nest box with landing board, perches, platform, and a ramp leading to the floor. The test pen contained vertical poles which held the landing perch during testing and a dummy rewarding apparatus for habituation.

**Figure 2 animals-10-01223-f002:**
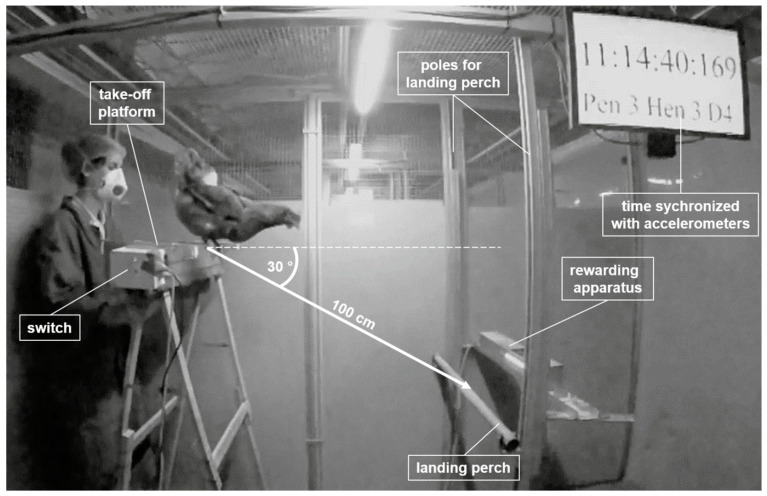
Testing apparatus in the test pen. Landing perch and rewarding apparatus are installed on both sides of the vertical poles. In this example, the take-off platform is placed with 100 cm diagonal distance to the landing perch with a 30° angle. The switch to open the flap of the rewarding apparatus is mounted on the take-off platform. The experimenter has her hands placed underneath the take-off platform with a neutral body posture.

**Figure 3 animals-10-01223-f003:**
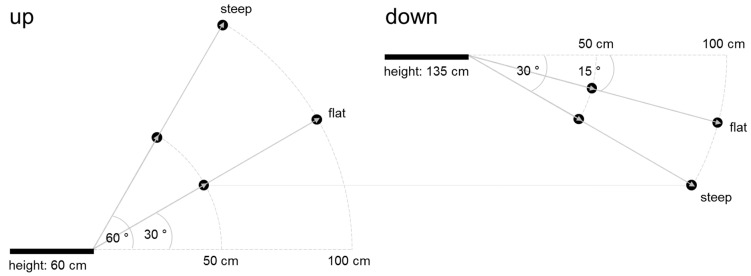
Test conditions based on a 2 × 2 × 2 factorial design combining steep and flat angles, 50 cm and 100 cm distance, and upward and downward jumps. The bar represents the take-off platform at different heights for up and down, whereas the dots show the positioning of the landing perch for all treatment combinations. The landing perch for the condition up/50 cm/flat was installed at the same height as the landing perch for the condition down/100 cm/steep.

**Figure 4 animals-10-01223-f004:**
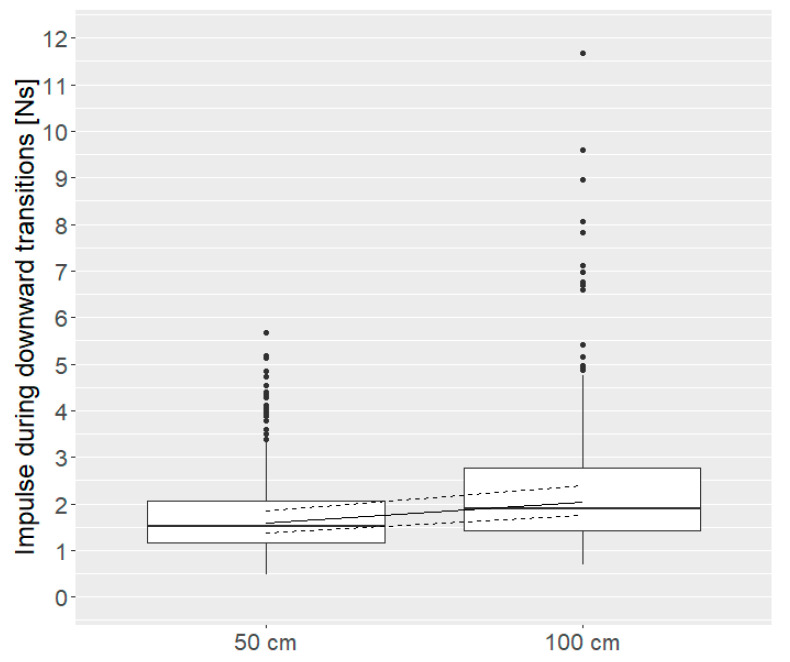
Impulse during downward landing (Ns) for 50 cm and 100 cm distance (distance: *p* = 0.001). Boxes show medians, lower and upper interquartile range. Whiskers indicate 1.5 times the interquartile range. In addition, model estimates with estimated means (solid line) and 95 % confidence intervals (dashed lines) are shown.

**Figure 5 animals-10-01223-f005:**
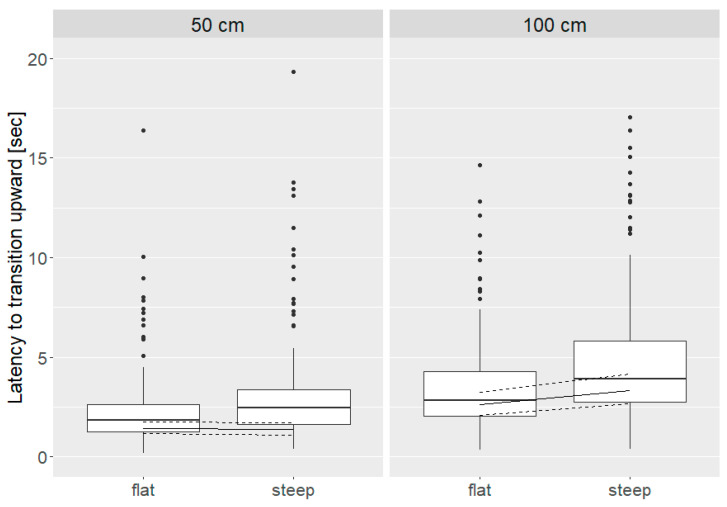
Latency to transition (sec) for distances of 50 cm and 100 cm and flat and steep angles (distance * angle: *p* = 0.048). Boxes show medians, lower and upper interquartile range. Whiskers indicate 1.5 times the interquartile range. In addition, model estimates with estimated means (solid line) and 95% confidence intervals (dashed lines) are shown.

**Figure 6 animals-10-01223-f006:**
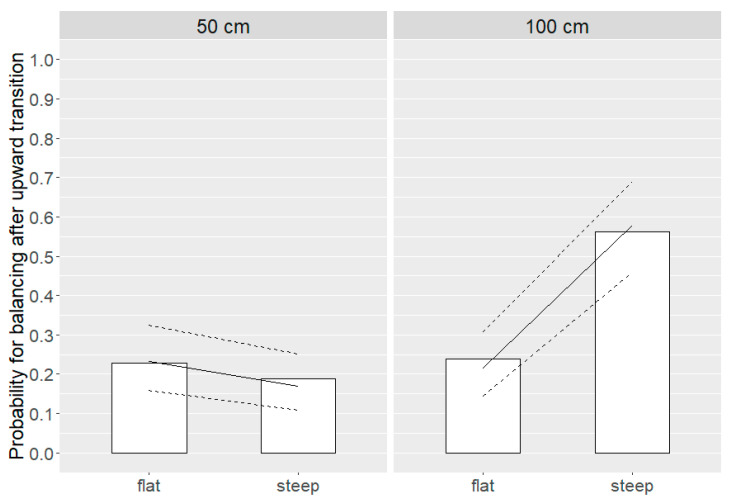
Probability for balancing movements after upward transitions of 50 cm and 100 cm distance and flat and steep angles (distance * angle: *p* = 0.0001). Bars represent the ratio of hens performing balancing movements after transitions of a specific condition. In addition, model estimates with estimated means (solid line) and 95 % confidence intervals (dashed lines) are shown.

**Figure 7 animals-10-01223-f007:**
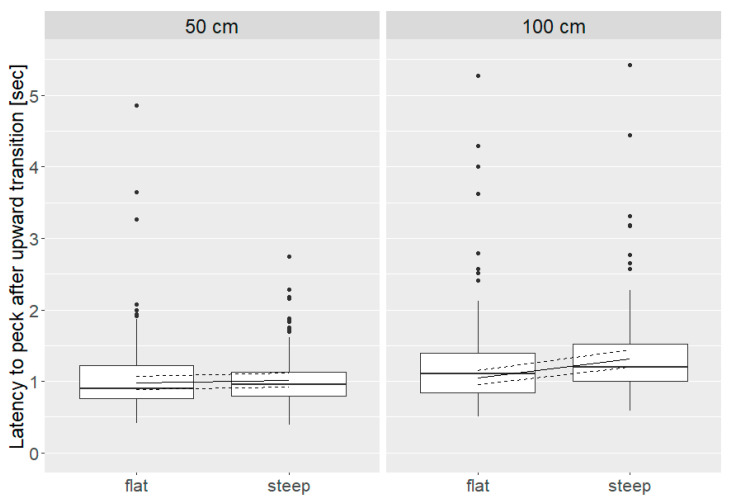
Latency to peck (sec) after upward transitions of 50 cm and 100 cm distance and flat and steep angle transitions (distance * angle: *p* = 0.001). Boxes show medians, lower and upper interquartile range. Whiskers indicate 1.5 times the interquartile range. In addition, model estimates with estimated means (solid line) and 95 % confidence intervals (dashed lines) are shown.

**Table 1 animals-10-01223-t001:** Variables recorded from video.

Variable		Description
Signal	hh:mm:ss.000	LED on
Take-off start	hh:mm:ss.000	first forward/upward movement (upward jumps) or forward/downward movement (downward jumps)
Take-off end	hh:mm:ss.000	both feet (and toes) lose contact to take-off platform
Landing start	hh:mm:ss.000	first contact with the landing perch
Landing end	hh:mm:ss.000	both feet on the perch and body of the hen stabilized with both wings folded close to the body, head pointing forward, and tail in a horizontal position
First peck	hh:mm:ss.000	first peck directed towards the reward indicated by head reaching the lowest position and neck feathers erecting

**Table 2 animals-10-01223-t002:** Overview of results including outcome variables, the variables remaining in the model after stepwise-backwards reduction (Effects), *p*-values for each variable or interaction, as well as the pattern in the results.

Outcome Variable			Effects ^1^	*p*-Value	Pattern
Peak force	Upward	Take-off	Distance	0.027	100 cm > 50 cm
		Flight	Distance * angle	0.037	50 cm: flat > steep 100 cm: flat = steep
		Landing	Distance	0.019	100 cm > 50 cm
	Downward	Take-off	Hybrid	0.032	Brown Nick > Nick Chick
		Flight	Hybrid	0.008	Brown Nick > Nick Chick
			Distance * angle	0.038	50 cm: flat = steep 100 cm: steep > flat
		Landing	Distance	0.019	100 cm > 50 cm
			Hybrid * angle	0.026	Nick Chick: flat = steep Brown Nick: steep > flat Brown Nick > Nick Chick
Impulse	Upward	Take-off	Distance * angle	0.036	50 cm: flat > steep 100 cm: flat < steep
		Landing	Distance * angle	0.001	50 cm: flat = steep 100 cm: steep > flat 100 cm > 50 cm
	Downward	Take-off	Hybrid	0.018	Brown Nick > Nick Chick
			Distance * angle	0.004	50 cm: flat = steep 100 cm: steep > flat
		Landing	Distance	0.001	100 cm > 50 cm
Latency to transition	Upward		Hybrid	0.016	Brown Nick > Nick Chick
			Distance * angle	0.048	50 cm: flat = steep 100 cm: steep > flat 100 cm > 50 cm
	Downward		Distance	0.001	100 cm > 50 cm
			Angle	0.001	Steep > flat
Balancing movements	Upward		Distance * angle	0.001	50 cm: flat = steep 100 cm: steep > flat
	Downward		Distance	0.0001	100 cm > 50 cm
Latency to first peck	Upward		Distance * angle	0.001	50 cm: flat = steep 100 cm: steep > flat
	Downward		Hybrid	0.034	Nick Chick > Brown Nick
			Distance * angle	0.006	50 cm: flat = steep 100 cm: steep > flat 100 cm > 50 cm

^1^ Variables remaining in the model after stepwise-backwards reduction.

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
