# Peer review of "Perch Positioning Affects both Laying Hen Locomotion and Forces Experienced at the Keel"

_animals, 2020, doi:10.3390/ani10071223_

Round 1

Reviewer 1 Report

Dear authors,

I've read your manuscript with much interest. The topic of keel bone damage is a relatively new one, but the importance for animal welfare is becoming clearer and clearer. This study aids in gathering more knowledge on this topic by providing more insight into the forces put on the keel bone. 

Overall, the manuscript was of good quality. I think the Results section can be improved by adding figures/tables providing a better overview. In addition, data on falls or collisions in relation to the testing situation (distance, angle, genetic strain) would be interesting to include.

Below you can find some more specific comments.

===============

L72: it’d be interesting to briefly add the characteristics of this system that are relevant for this study, i.e. the perch characteristics (height, positioning, angle, …). Also for the perches in the wintergarden mentioned in L74. This is relevant because the type  of perch these hens were reared with, might affect which perch type they cope with best later in life.

L86: ‘prevent bias’ – how do you know the hens have not already developed a bias in the rearing system?

L93-94: did you monitor whether hens always used the ramps?

L108: where were these windows positioned?

L127: please add the exclusion criteria. L495-497: these criteria are not very specific but seem quite subjective. Also, it seems a risk to exclude birds with a certain coping style, while e.g. hens with an inactive coping style may also not have been habituated to the test area, but were selected as focal hen. Did you try to prevent this bias?

L137: please include the possible positions of the take-off platform in Figure 1.

L161: is distance the horizontal or diagonal distance between the platform and perch?

L181-183: how was this information used? I don’t see this in the statistical analysis or results. Was information on keel bone damage also collected after the testing period and related to the results presented here?

Throughout Results section: when there is a significant interaction, please provide the P levels of the pairwise interactions as well.

The results section contains a lot of information in text, and only some of the results are shown in Figures. More figures would be helpful for an easier understanding of the results, and/or perhaps include a table with an overview of the results.

L387-389: interesting finding. This may also mean that lighting conditions can play a crucial role in successful transitions. Have there been any studies on this? Perhaps it’d also be good to describe the lighting conditions in this study in a bit more detail (spectrum, lux, …).

L411: you talk about unsafe landing resulting in greater loads at the keel bone – how would you define these – merely by the greater load or can it also be derived from behavioural observation? Did you observe whether landings were ‘safe’ or ‘unsafe’ in this study, and under which testing conditions they occurred more/less? L417-418: apparently falls and collisions were recorded, is it possible to present these data as well?

L413-414: is it necessary to determine an upper threshold? I can also imagine that lower forces, but experienced repeatedly, can have a similar or even worse effect than one high-force collision. How would you use the information on the upper threshold?

L424: define greenstick fractures

L448: can you give a range of these angles and distances encountered in common commercial systems?

L454: ‘in order to reduce keel bone fracture occurrence’ – it’d be more correct to say ‘in order to reduce the load on the keel bone which could reduce fracture occurrence’, because the occurrence of fractures has not been tested in this study.

L454-456: do you think this is feasible under commercial settings? If there are multiple rows of aviaries, the distance between them will always be more than 50 cm. Would it also be an option to provide ramps? Or, place the rows further apart – how far apart would they need to be in order to prevent hens jumping from one row to the other? Could be an interesting research question.

L490: was there a reason to catch the birds by the legs instead of holding them under the breast / around the wings? This so-called Swedish method is less stressful for the hens.

L513: highest rank per pen = the birds that were on average isolated in the test pen the fastest?

L540/547: sentence is incorrect

L561: were falls recorded in the testing phase, or did they not occur? If so, why are these data not presented?

Author Response

Reviewer #1

Dear authors,

I've read your manuscript with much interest. The topic of keel bone damage is a relatively new one, but the importance for animal welfare is becoming clearer and clearer. This study aids in gathering more knowledge on this topic by providing more insight into the forces put on the keel bone. 

Overall, the manuscript was of good quality. I think the Results section can be improved by adding figures/tables providing a better overview. In addition, data on falls or collisions in relation to the testing situation (distance, angle, genetic strain) would be interesting to include.

Below you can find some more specific comments.

### Thank you for your kind feedback and the valuable input. We are glad that you see our manuscript as a contribution to the field. Please find more detailed comments below.

===============

L72: it’d be interesting to briefly add the characteristics of this system that are relevant for this study, i.e. the perch characteristics (height, positioning, angle, …). Also for the perches in the wintergarden mentioned in L74. This is relevant because the type  of perch these hens were reared with, might affect which perch type they cope with best later in life.

### Thank you for pointing out the importance of rearing system features. We agree that these might affect the hens’ behavior later in life and added information about perch characteristics and positioning in the house and the wintergarden. We added the information about perches of the rearing system in lines 78-84.

L86: ‘prevent bias’ – how do you know the hens have not already developed a bias in the rearing system?

### Thank you for this comment. The hens could have developed a bias towards certain angles / distances during rearing, which could not be avoided logistically. Here, we mean preventing bias in the layer system. We added text to clarify (L97).

L93-94: did you monitor whether hens always used the ramps?

### We monitored ramp use during the first few days to make sure that all hens could access the perches. However, we did not monitor the ramp use after this point, and some hens would still jump from the perches to the litter without using the ramps. We added text to include the possibility that hens could still jump to the litter directly (L106-107).

L108: where were these windows positioned?

### The windows were incorporated in the walls of the building, approximately at 2 m distance from the short side of the pens. We added this information to the text (L122-123).

L127: please add the exclusion criteria. L495-497: these criteria are not very specific but seem quite subjective. Also, it seems a risk to exclude birds with a certain coping style, while e.g. hens with an inactive coping style may also not have been habituated to the test area, but were selected as focal hen. Did you try to prevent this bias?

### Thank you for this comment. We agree that these criteria neglected that hens with an inactive coping style might have been selected as focal hens despite a lack of habituation to the testing pen. Nevertheless, the selection of focal hens was continued over a period of multiple weeks (see Appendix A and B), and hens not coping with the situation would have been excluded in subsequent training steps. This was indeed the case, as only 40 out of the 80 initially selected hens could be fully trained.

L137: please include the possible positions of the take-off platform in Figure 1.

### Figure 1 displays the pen set-up during periods of non-training and non-testing day, i.e., when hens had access to both the test and the home pen. Adding the take-off platform to this graph might imply that the platform was always present in the test pen. We added the information which doors were open or closed in this set-up and added text to the legend to explain that this was not the test set-up (L117). We also added schematics of the set-up in the test pen during training and testing (new Figure 2).

L161: is distance the horizontal or diagonal distance between the platform and perch?

### We added “diagonal” to clarify (L100, L173, L182). The distances are also shown in Figure 3 (L192).

L181-183: how was this information used? I don’t see this in the statistical analysis or results. Was information on keel bone damage also collected after the testing period and related to the results presented here?

### Thanks for raising this question. Keel bone condition was assessed for each individual bird before each testing round while installing the accelerometer. We intended to exclude hens with fractures or foot pad conditions, but no fractures were detected using palpation. We added this information (L204).

Throughout Results section: when there is a significant interaction, please provide the P levels of the pairwise interactions as well.

### We did not calculate pairwise comparisons due to multiple testing issues and because post hoc tests increase the risk of false positives. In addition, as we did not have hypotheses for specific pairwise comparisons, post hoc tests would have resulted in decreased power. Instead, we decided to present the modelled means and confidence intervals, which give the reader the chance to assess the biological relevance of differences found between treatment groups in interactions. We followed the suggestions by Nakagawa 2004 (https://academic.oup.com/beheco/article/15/6/1044/206216).

The results section contains a lot of information in text, and only some of the results are shown in Figures. More figures would be helpful for an easier understanding of the results, and/or perhaps include a table with an overview of the results.

### Thank you for these suggestions. We added a table summarizing all the results (Table 2, L311), but decided not to add more figures as the results are so numerous.

L387-389: interesting finding. This may also mean that lighting conditions can play a crucial role in successful transitions. Have there been any studies on this? Perhaps it’d also be good to describe the lighting conditions in this study in a bit more detail (spectrum, lux, …).

### Thank you for this comment. We agree that lighting conditions can play a crucial role in successful transitions, and that this might be especially relevant in aviary systems where different tiers might have different exposures to light sources. We discuss the importance of lighting during dawn and dusk on transition accuracy further below in the discussion and added a text to describe the risk for falls in low-light conditions (L311).

L411: you talk about unsafe landing resulting in greater loads at the keel bone – how would you define these – merely by the greater load or can it also be derived from behavioural observation? Did you observe whether landings were ‘safe’ or ‘unsafe’ in this study, and under which testing conditions they occurred more/less? L417-418: apparently falls and collisions were recorded, is it possible to present these data as well?

### Thank you for this question. Unsafe landings were defined based on behavior exclusively but were found to correlate with high impacts in later studies, e.g., Baker et al. 2020. For instance, Stratmann et al. (2015) described a successful landing phase as “the hen’s feet contacting the ground at the same time with an upright body position”. We adapted this definition as we also included landings followed by balancing movements or asymmetric landings as “safe”. On the other hand, collisions and falls were classified as “unsafe”. We added this information in our Material & Methods section (L228-231). We also recorded unsafe landings (i.e., falls and collisions) during testing. However, as the aim of this study was to investigate forces at the keel during controlled transitions, the hens were carefully trained in order to avoid unsafe landings to the greatest extent possible. Very few landings during the testing phase were unsafe, and thus, these data were not presented. In the supplementary material, unsafe landings are represented by missing data points (e.g., the data entry for jump number 1 is missing for hen 5, pen 1, down / flat / 100 cm, which means that this transition resulted in an unsafe landing).

L413-414: is it necessary to determine an upper threshold? I can also imagine that lower forces, but experienced repeatedly, can have a similar or even worse effect than one high-force collision. How would you use the information on the upper threshold?

### Thank you for pointing this out. We agree that lower forces applied repeatedly can lead to fractures. Here, we mean an upper threshold for traumatic events which result in fractures (e.g., when a hen collides with a perch). We added text to clarify (L438).

L424: define greenstick fractures

### We defined greenstick fractures as incomplete, bending fractures commonly found in developing bone (L449).

L448: can you give a range of these angles and distances encountered in common commercial systems?

### We added the maximum diagonal distance and angle found in Swiss aviary systems and added a reference (Stratmann & Ringgenberg 2018; L477-478).

L454: ‘in order to reduce keel bone fracture occurrence’ – it’d be more correct to say ‘in order to reduce the load on the keel bone which could reduce fracture occurrence’, because the occurrence of fractures has not been tested in this study.

### Thank you for this clarification. We rephrased this sentence according to your suggestion (L483).

L454-456: do you think this is feasible under commercial settings? If there are multiple rows of aviaries, the distance between them will always be more than 50 cm. Would it also be an option to provide ramps? Or, place the rows further apart – how far apart would they need to be in order to prevent hens jumping from one row to the other? Could be an interesting research question.

### Thank you for raising this question. The feasibility under commercial conditions is often discussed and placing the aviary rows further apart is often not an option – even though a study has shown that the distance between tiers can be a risk factor for keel bone fractures (Heerkens et al. 2016). We added this information (L487-488). We mention ramps as an option to improve the housing system in the last paragraph of the discussion, as ramps are indeed a good option to facilitate transitions between rows and tiers.

L490: was there a reason to catch the birds by the legs instead of holding them under the breast / around the wings? This so-called Swedish method is less stressful for the hens.

### Thanks for the clarification. The Swedish method was indeed used for carrying the hens – grabbing them by both legs was only used for the initial catching. We added this information to the Appendix (L521).

L513: highest rank per pen = the birds that were on average isolated in the test pen the fastest?

### Yes, correct. We added this information in parentheses for clarification (L544-545).

L540/547: sentence is incorrect

### Thank you for picking this up. We corrected the sentence (L572, L579).

L561: were falls recorded in the testing phase, or did they not occur? If so, why are these data not presented?

### Please see comment above: Falls were recorded but occurred very rarely.

Reviewer 2 Report

The manuscript of Rufener et al is an interesting contribution to the field of housing systems for laying hens. Plagiarism check showed no overlap with previously published papers.. In addition, the results are novel and the methodology used, especially the statistical analysis is correct..I recommend to remove the subsections from the discussion and add the limitations. Following that I recommend this paper for publication. 

Author Response

Reviewer #2

The manuscript of Rufener et al is an interesting contribution to the field of housing systems for laying hens. Plagiarism check showed no overlap with previously published papers.. In addition, the results are novel and the methodology used, especially the statistical analysis is correct..I recommend to remove the subsections from the discussion and add the limitations. Following that I recommend this paper for publication. 

### Thank you for your positive feedback. We removed the subsections in the discussion. Unfortunately, we are not sure what you mean with adding the limitations. We had mentioned the limitations of our study in lines 435-438, 441-442, 453-455, 471-474, and 474-479 , hoping that this fulfills your suggestion.

Reviewer 3 Report

Overall

I find the paper to be well written on an important topic. The set up of the study is intriguing and novel. I have some comments to the paper, see below.

Title

The title does not flow easily. I had to read it several times to get the meaning of “forces experienced at the keel”. A suggestion is to add “both”: Perch positioning affects both laying hen locomotion and forces experienced at the keel.

Summary

Yes, collisions and falls are one of many explanatory factors for KBF. However, there are many more. Suggest you add a sentence that highlights this.

“In addition, routine behaviors such as dustbathing or locomotion could contribute to the problem due to accumulated forces at the keel.” Is this documented or a mere speculation? Or is this the hypothesis of the paper?

Abstract

Please add the age when the birds were tested.

Introduction

“…the height and difficulty maneuvering within the system likely increase the risk for falls and injuries.” Is this documented or a mere speculation? Or is this the hypothesis of the paper?

Line 41: One of the reasons….

Line 46: “Although falls and collisions are suspected to result in fractures”, please, add reference.

Line 47-48: “…cannot be the only…”. I do believe Thøfner et al actually says that collisions are not at all a likely explanation.

Line 61: please add anatomically position at the keel bone for the placement of the accelerometer.

Animals, Material and Methods

Was the barn a research facility or a commercial farm?

What was the stocking density before the birds were given outdoor access?

Line 182: were hens with a positive palpation score included in the study? was palpation the only method to evaluate keel bones?

Results

Was the latency to transition affected by whether or not the hen had a keel bone fracture?

Please add this information, along with the KBF prevalence at the time of testing.

Discussion

Due to the high numbers of KBF reported elsewhere I assume that a lot of the hens in your study would have fractures. Therefore, I find it a bit strange that you do not report the prevalence of KBF in your study.  I assume that this, on hen level, most definitely would affect your transition results, along with latency to peck. You should discuss this.

Line 379-381: can you be sure that this is due to the steepness and not due to reluctance to move due to fracture/pain? You should discuss this. Likewise, for line 394-395.

Author Response

Reviewer #3

Overall

I find the paper to be well written on an important topic. The set up of the study is intriguing and novel. I have some comments to the paper, see below.

### Thank you for your positive feedback and your valuable input. Please find the detailed answers to your comments below.

Title

The title does not flow easily. I had to read it several times to get the meaning of “forces experienced at the keel”. A suggestion is to add “both”: Perch positioning affects both laying hen locomotion and forces experienced at the keel.

### Thank you for this suggestion. We agree that adding “both” improves the flow of the title, and changed it accordingly (L2).

Summary

Yes, collisions and falls are one of many explanatory factors for KBF. However, there are many more. Suggest you add a sentence that highlights this.

### Thank you for this comment. We added text to point out that KBF are a multifactorial problem (L12-13). 

“In addition, routine behaviors such as dustbathing or locomotion could contribute to the problem due to accumulated forces at the keel.” Is this documented or a mere speculation? Or is this the hypothesis of the paper?

### It has been suggested that routine behaviors could result in fractures, e.g., by Harlander-Matauschek et al. 2005, but this aspect of KBF has not been studied. We slightly rephrased this sentence (L14). However, as this is the simple summary which should be understandable for lay readers and because we discuss routine behaviors as a potential cause for fractures in greater detail in the introduction and discussion, we prefer to avoid going into more detail here.

Abstract

Please add the age when the birds were tested.

### Thank you for picking this up. We added the age of the birds (L28).

Introduction

“…the height and difficulty maneuvering within the system likely increase the risk for falls and injuries.” Is this documented or a mere speculation? Or is this the hypothesis of the paper?

### Thank you for this comment. There has been one single study linking system height with KBF prevalence. We added text to point out that this relationship is an assumption which is based on findings of one study only (L41-43). In addition, we added the corresponding reference (Wilkins et al. 2011) as well as a review article which discusses this general assumption (Rufener & Makagon, in press).

Line 41: One of the reasons….

### Thank you for pointing out the multifactorial nature of KBF. We adapted the text according to your suggestion (L43-44).

Line 46: “Although falls and collisions are suspected to result in fractures”, please, add reference.

### We added Stratmann et al. 2015 as a reference (L48).

Line 47-48: “…cannot be the only…”. I do believe Thøfner et al actually says that collisions are not at all a likely explanation.

### Thank you for pointing this out. We revisited Thøfner et al., who state that “high energy collisions cannot be responsible for the majority of fractures”. Some of the fractures in their study showed markers associated with trauma, which suggest that collisions seem to be the cause in a few cases only. We rephrased this sentence accordingly (L49-50).

Line 61: please add anatomically position at the keel bone for the placement of the accelerometer.

### We added text to clarify where the accelerometer was placed (L63-64).

Animals, Material and Methods

Was the barn a research facility or a commercial farm?

### This study was conducted in a research facility. We added text to clarify (L90).

What was the stocking density before the birds were given outdoor access?

### The indoor stocking density was 16.4 birds per m2 grid surface. We added this information to the text (L84).

Line 182: were hens with a positive palpation score included in the study? was palpation the only method to evaluate keel bones?

### We palpated all hens before each testing session in order to exclude them from the study if keel bone fractures were found. However, we did not find any fractures based on palpation. We added this information to the text (L204). For this study, palpation was the sole method of fracture detection.

Results

Was the latency to transition affected by whether or not the hen had a keel bone fracture?
Please add this information, along with the KBF prevalence at the time of testing.

### Thanks for this question. As stated above, none of the hens had fractures based on palpation. We added this information (L204).

Discussion

Due to the high numbers of KBF reported elsewhere I assume that a lot of the hens in your study would have fractures. Therefore, I find it a bit strange that you do not report the prevalence of KBF in your study.  I assume that this, on hen level, most definitely would affect your transition results, along with latency to peck. You should discuss this.

### Thank you for your comment. We agree that KBF are likely to affect mobility behavior and thus, our results. Nevertheless, as mentioned above, none of the hens included in the study had fractures based on palpation. We started training and testing the hens as early as possible in order to avoid fractures – the hens were moved from the rearing barn to the layer barn at 16 weeks of age instead of 19 weeks of age, as it is usually done at this facility. We assume that the provision of a ramp, the relatively low age, as well as low animal density in the pens contributed to a later onset of fractures. We added text to discuss the importance of fractures for mobility and possible effects transitions in a commercial environment (L471-474).

Line 379-381: can you be sure that this is due to the steepness and not due to reluctance to move due to fracture/pain? You should discuss this. Likewise, for line 394-395.

### Thank you for your question. We agree that the effect of perch positioning needs to be disentangled from reluctance to move due to a fracture. As none of our focal hens had fractures, we are indeed able to say that longer latency was due to perch positioning. We added the information about keel integrity of the focal birds in the manuscript (L204).